# Superparamagnetic Iron Oxide Nanoparticles—Current and Prospective Medical Applications

**DOI:** 10.3390/ma12040617

**Published:** 2019-02-19

**Authors:** Joanna Dulińska-Litewka, Agnieszka Łazarczyk, Przemysław Hałubiec, Oskar Szafrański, Karolina Karnas, Anna Karewicz

**Affiliations:** 1Chair of Medical Biochemistry, Jagiellonian University Medical College, 7 Kopernika St., 31-034 Kraków, Poland; agnieszka013@neostrada.pl (A.Ł.); przemyslawhalubiec@gmail.com (P.H.); osk.sza2@gmail.com (O.S.); 2Department of Chemistry, Jagiellonian University, 2 Gronostajowa St., 30-387 Kraków, Poland; karolina.karnas@student.uj.edu.pl (K.K.); karewicz@chemia.uj.edu.pl (A.K.)

**Keywords:** SPION, MRI, hyperthermia, iron oxide, antibodies, toxicity

## Abstract

The recent, fast development of nanotechnology is reflected in the medical sciences. Superparamagnetic Iron Oxide Nanoparticles (SPIONs) are an excellent example. Thanks to their superparamagnetic properties, SPIONs have found application in Magnetic Resonance Imaging (MRI) and magnetic hyperthermia. Unlike bulk iron, SPIONs do not have remnant magnetization in the absence of the external magnetic field; therefore, a precise remote control over their action is possible. This makes them also useful as a component of the advanced drug delivery systems. Due to their easy synthesis, biocompatibility, multifunctionality, and possibility of further surface modification with various chemical agents, SPIONs could support many fields of medicine. SPIONs have also some disadvantages, such as their high uptake by macrophages. Nevertheless, based on the ongoing studies, they seem to be very promising in oncological therapy (especially in the brain, breast, prostate, and pancreatic tumors). The main goal of our paper is, therefore, to present the basic properties of SPIONs, to discuss their current role in medicine, and to review their applications in order to inspire future developments of new, improved SPION systems.

## 1. Introduction 

An idea that the primordial cause of the majority of known diseases is located at the molecular level stimulates a rising interest in applying nanotechnology to explore further this possibility. Working in nanoscale could give us a proper way to diagnose, treat, or even prevent numerous pathological conditions. Among the solutions which are tested worldwide in laboratories and which are starting to be introduced into hospitals, SPIONs (Superparamagnetic Iron Oxide Nanoparticles) emerge with great potential.

In general, SPIONs are particles formed by small crystals of iron oxide (commonly called magnetite Fe_3_O_4_ or maghemite γ-Fe_2_O_3_), which may be surface-modified to gain colloidal stability in aqueous media. These modifications may involve capping SPIONS with organic acids (e.g., citric acid [1]) or coating them with a biocompatible, hydrophilic polymer (e.g., poly(ethylene glycol) [2]), or polysaccharides [3].

The nanometric size (usually in the range of 20 to 150 nm) and sensitivity to a magnetic field (Figure 1) make SPIONs unique. As they might be conjugated with various molecules, a range of possibilities of their use is wide. A popular solution is to bind antitumor antibodies to SPIONs’ surfaces and inject the resulting, targeted particles into the circulatory system [4,5]. They could be used then to track the tumor cells (as they can be detected with MRI—Magnetic Resonance Imaging) or to kill them (through releasing drugs or by magnetic hyperthermia). Another interesting application is to use SPIONs in diagnosing and further monitoring early stages of endothelial inflammation, one of the early symptoms of cardiovascular diseases [6]. However, despite the fact that basic rules and ideas regarding SPIONs seem to be simple, an appropriate understanding and introducing them into clinical practice require a more detailed insight. Both the molecular structure of SPIONs and the conditions inside a human body have to be considered. For example, iron ions catalyze the well-known Haber–Weiss and Fenton reactions, which are responsible for generation of ROS (Reactive Oxygen Species) and, therefore, could induce cell-damage in the healthy cells. The influence of the coating of iron oxide core in SPION on its critical properties, such as cellular uptake or total clearance, is also remarkable. The contrasting increase observed in MRI images from SPIONs in blood and those which were phagocytized by liver macrophages is different. The influence of bloodstream velocity and vessel diameter on the SPIONs’ performance as an MRI contrast agent should also be considered.

In the last few years, the understanding of all the complex relations mentioned above increased significantly, improving the clinical efficiency of SPIONs. The main goal of the present review is to sum up the up-to-date knowledge on SPIONs and to present the most promising examples of their clinical applications. We will first take a closer look at SPION properties, focusing on their coatings. Then we will compare and discuss the four most frequently investigated clinical usages of SPIONs—binding antibodies (1), acting as a MRI contrast agent (2), magnetic hyperthermia (3), and drug delivery (4)—as well as the correlations between them. Next, we will show new directions of SPION development. Finally, their limitations and side effects, as well as possible ways to prevent them, will be presented.

## 2. Physicochemical Properties of SPIONs

As mentioned above, SPION generally consists of an iron oxide core and coating derived from organic compounds.

There are three types of iron oxides that may form the core of a SPION—magnetite (Fe_3_O_4_), maghemite (γ-Fe_2_O_3_) and hematite (α-Fe_2_O_3_). Their critical properties, such as molecule size and magnetic parameters, are very different [7].

Magnetite is predominantly chosen among the abovementioned iron oxides. It contains Fe^2+^ and Fe^3+^ ions in the 1:2 ratio. It is important, as Fe^2+^ triggers the Fenton reaction resulting in ROS production in the cells.

Magnetite in its natural condition is considered to be ferrimagnetic (rarely antiferromagnetic). It is said to have the strongest magnetic properties of all the transition metal oxides [8]. After ferrimagnetic material is subjected to the external magnetic field, it stays magnetized to some extent even when the field is removed. The reason is that the magnetic moments stay aligned even in the absence of a magnetic field. As a result, their magnetization moment M shows a hysteresis loop in the function of external magnetic field H. However, Fe_3_O_4_ crystals with diameters of 20 nm or less are superparamagnetic (there is no hysteresis after applying an external magnetic field because each crystal acts as a single magnetic domain). In other words, SPIONs reveal their magnetic properties only when subjected to an external magnetic field, whereas bulk magnetite preserves magnetic properties also in the absence of an external magnetic field. Maghemite and hematite both contain Fe^3+^ ions and, despite that in bulk they have different magnetic properties (ferromagnetic and antiferromagnetic, respectively), after forming sufficiently small crystals, they both turn superparamagnetic. Hematite (α-Fe_2_O_3_) nanoparticles are successfully synthesized by using the hydrothermal synthesis method. An X-ray powder diffraction (XRPD) of the sample shows the formation of the nanocrystalline α-Fe_2_O_3_ phase. Transmission electron microscopy (TEM) measurements show the spherical morphology of the hematite nanoparticles and the narrow size distribution. The magnetic properties were measured using a superconducting quantum interference device (SQUID) magnetometry. An investigation of the magnetic properties of hematite nanoparticles demonstrated a divergence between the field-cooled (FC) and zero-field-cooled (ZFC) magnetization curves shown below Tirr = 103 K (irreversibility temperature). The M(H) (magnetization versus magnetic field) dependence at 300 K showed the properties of the SPION. Furthermore, magnetic measurements showed a high magnetization at room temperature (M_w_ = 159.687 g/mol) which is desirable for application in spintronics and biomedicine. Finally, the core–shell structure of the nanoparticles was used to describe the high magnetization of the hematite nanoparticles [9]. Many parameters of SPIONs are defined by their external layer. Without the coating, SPIONs tend to aggregate; they also are hydrophobic and, when injected into the bloodstream, are coated by plasma proteins (the so-called opsonization). Hydrophilic coating can prevent or significantly decrease opsonization, and through electrostatic interactions or steric hindrance, it decreases SPION aggregation. The surface charge of the coating is essential. Zeta potential, which represents a surface charge of the particles in a colloidal suspension, is one of the most important factors defining their stability, tendency to aggregate (thus defining their effective size), as well as their ability to bind serum proteins. In spite of the fact that most of them are charged negatively, the more positive the charge of a SPION is, the stronger its ability to bind serum proteins [10].

The coating can prevent the release of iron ions and can decide about the nanoparticle’s interactions with its biological environment (cells, proteins, etc.); therefore, it strongly affects their toxicity. To increase their biocompatibility, as well as to target them to specific cells or tissues, the surface of SPIONs can be modified by attaching various functional groups, as well as ligands or antibodies [11] Both the size and hydrophilicity of surface modifying agents are also vital for the process of cellular uptake and the inside-cell fate of SPION. When the particle radius exceeds 100 nm, it starts to be strongly phagocytosed, while those which are under 30 nm are quickly uptaken by pinocytosis. A higher hydrophilicity reduces pinocytosis by preventing a particle crossing through the hydrophobic, lipid bilayer of the cells. 

In recent studies, the most commonly applied coating agents are polymers: PEG (Poly(Ethylene Glycol)) (1), PVA (Poly(Vinyl Alcohol)) (2), chitosan (3), PVP (Poly(Vinyl Pyrrolidine)) (4), and dextran (5) (Figure 2).

(1) PEG is characterized by a good biocompatibility as it carries no potentially toxic groups. Its structure [(–CH_2_–CH_2_–O–)_n_] indicates possible interactions with alcohol dehydrogenase and the production of acetaldehyde as a result. Such an effect has been confirmed; however, it strongly depends on the chain length and occurs only for PEG macromolecules with a molecular weight lower than 1000 Da (n ≈ 17) [12]. Hydroxyl groups at the chain ends allow for the attachment of antibodies and other agents to SPIONs. PEG was also recognized to improve half-life time and to reduce the cellular uptake of SPIONs [2].

(2) PVA is another polymeric agent with the structure similar to PEG [(–CH_2_–CH(OH)–)_n_]. It is reported to be highly biocompatible because PVA-SPIONs are scarcely captured by the cells (if there has not been a magnetic field applied) [13]. Highly beneficial is also their ability to prevent NPs (Nanoparticles) clumping inside particularly narrow vessels of 7–8 µm in diameter, which is reported to impede SPION usage [14].

(3) Chitosan is derived from chitin, a naturally occurring polysaccharide, by deacetylation. Its chain is built from β-1,4-D-glucosamine and N-acetyl-D-glucosamine units, their ratio being defined by the deacetylation degree. Chitosan is a highly biocompatible, hydrophilic, and cationic polymer. The ease of conjunction of thiol groups (–SH) with chitosan allows it for binding mucous glycoproteins very effectively (as they usually have one cysteine-rich domain on each end of their chain), showing some antiprotease activity (through chelating zinc ions, which are often necessary for proteases activity) or regenerating glutathione dimers (GSSG) [10]. SPIONs coated with the cationic trimethylammonium derivative of chitosan were reported to increase the number of inflammatory cells (in mice) [15]; however, it is possible to modify chitosan with anionic groups (carboxylic, sulphonic, or phosphate), which could decrease the chitosan tendency to induce the abovementioned unwanted changes while retaining its favorable properties.

The half-life time of SPIONs coated with both (4) PVP and (5) dextran in circulation is prolonged; due to their improved colloidal stability, PVP-SPIONs did not affect cell viability in a significant manner until a high concentration of 250 μg/mL was used [16], while Singh et al. [17] reported that dextran-coated SPIONs were cytotoxic already at the concentration as high as 50 μg/mL. It was suggested that dextran coating is disrupted during interactions with the cell membrane or is digested by lysosomal dextranases. Iron oxide is then freely released into the cell, where it causes a toxic effect. Unterweger et al. [18] have, however, clearly shown that dextran-coated SPION formulations can be nontoxic even at concentrations of 400 µg/mL, which disagree with the previous findings. The authors suggested that the toxic effect exists due to the degradation of the dextran coating and due to the lack of stability of the dextran shell. They also suggested that not only the material but also the way in which it is bound to the particle’s surface plays a critical role regarding the stability and toxicity of dextran-coated SPIONs. 

Unterweger et al. [18] evaluated also an uptake of dextran-stabilized SPIONs (SPIONDex) in the THP-1 human monocytic cells (cell line derived from an acute monocytic leukemia patient), macrophages, and HUVECs (isolated and cultured in vitro in the 1970s). The cells were incubated with various particle concentrations (25–400 µg/cm^2^) for 24 h. The highest uptake was observed for the phagocytic macrophages, leading to an 11-fold (24 h) increase in the iron load per cell in comparison to the controls without particles. In the HUVECs and THP-1 cells, the uptake was far less pronounced.

Independently of their size, SPIONDex displayed no irritation potential in a chick chorioallantoic membrane assay. Cell uptake studies of ultrasmall (30 nm) SPIONDex confirmed their internalization by the macrophages but not by the non-phagocytic cells. Additionally, complement activation-related pseudoallergy (CARPA) experiments in pigs treated with ultrasmall SPIONDex indicated the absence of hypersensitivity reactions.

A thorough literature search shows that depending on the planned destination of NPs, different coatings may be used to maintain SPIONs in circulation for a longer or shorter time. It is also possible to manipulate the degree of cellular uptake of SPIONs or even the level of their toxicity toward selected cells/tissues (if carefully combined with reasonable targeting).

Iron oxide cores may be also coated with other types of agents. Such studies are reported in the literature, but their potential still remains unclear. For example, there was an idea to surround SPIONs with albumins [19,20] in order to avoid the uncontrolled natural attachment of serum proteins [21]. The main goal seemed to be reached [22], although it also resulted in a lower uptake and lysosomal decomposition of particles. However, those experiments were made in vitro or in simple organisms (nematodes *Caenorhabditis elegans*), and there emerges a question whether BSA (Bovine Serum Albumin) used as a shell in SPIONs will not cause a dangerous anaphylactic reaction in humans, as it is a popular food allergen [23].

Other potentially available shell agents (e.g., citrates, various proteins like HSP—Heat Shock Proteins—or inorganic compounds such as other metal oxides) will be discussed in the next sections, as their properties may be useful in some special clinical applications but not in the general practice, at least, according to the results of research carried out so far.

## 3. Summary of Clinical Applications of SPIONs

The essential properties of SPIONs were discussed above; now we are going to present the well-established as well as the recently proposed ways to exploit their possibilities in clinical practice. Our goal is to compare the results of different studies on a particular application of SPIONs, to point out the advantages and disadvantages of each approach, and to suggest further improvements. The main axis of the discussion would be the treatment of cancer diseases, as they are the second most likely cause of death in developed countries and seem to be a perfect target for techniques which engage SPIONs. Some alternative ways of using SPIONs will be also mentioned because a lot of them appear highly promising and inspiring for future investigations. 

### 3.1. SPIONs Conjugated with Antibodies

Antibodies (Ab) have already been used in various ways in oncological treatments (e.g., Herceptin in breast cancer) due to their ability to target specific patterns in proteins tertiary structure. They could be coupled with SPIONs to enhance their diagnostic and therapeutic value. As an increased specificity is vital for both diagnosis and treatment with SPIONs, the main ideas behind antibody-SPION systems will be discussed separately.

There are many reports of Abs covalently bound to SPIONs. The main restriction in this approach is the limited selectivity of antibodies (a problem that is well-known from any trial of using them in anticancer therapy). It could be overcome through the discovery of tumor-specific antigens (instead of tumor-associated antigens, which could also be present on other cells). SPIONs bound to Ab were not reported to affect the properties of antibodies in a significant way. On the other hand, Abs’ influence on SPIONs’ zeta potential, total hydrodynamic radius, and superficial energy is considerable.

There is a fine example of success in introducing such a solution. Pancreatic carcinoma is one of the deadliest tumors, and there is an urgent need to introduce a satisfying therapy. Mesothelin is a protein, which the membrane form was confirmed to be associated with a few tumors—pancreatic adenocarcinoma but also ovarian cancer or mesothelioma [24]. It was an inspiration for making in vivo studies on the effectiveness in targeting the pancreatic carcinoma cells in mouse xenografts by SPIONs coupled with anti-mesothelin antibodies [25]. The results were encouraging because of no relevant cytotoxicity and of a high specificity of this system to be an MRI contrast. The only organ that showed an increased storage of iron was liver.

It is a general problem that SPIONs tend to be absorbed by liver macrophages. However, this problem could be resolved by appropriately adjusting the particle’s shell—negatively charged PEG used as the coating was shown to increase the liver and spleen uptakes of SPIONs [26]. Chitosan as the coating material seems to prevent such an interaction [27].

The next example arises from the one of most strongly exploited directions of SPION usage in antitumor therapy. Some of them, e.g., with a PVA shell, showed an increased ability to cross the BBB (Blood–Brain Barrier) and to lower neurotoxicity [28], which indicates a new way to treat noninvasive brain tumors. Glioma is one of the most intensively studied here, as even such mainly noninvasive tumors seem to be deadly due to their location. Because there is a confirmed expression of mHSP70 (membrane Heat Shock Protein 70) on the glioma cells’ surface, although not on the healthy cells [29], the effectiveness of anti-cmHSP70 antibodies [30] in treating C6 glioma was studied. An interesting concept was to expose the tumor to a single, low dose of ionizing irradiation (10 Gy). This increased significantly the targeted protein’s expression on the tumor cells. Specific interactions of antibodies and their ligands were confirmed by MRI, as well as the further enhancement of sensitivity after irradiation. It is also worth noting that there is a novel approach of coating SPIONs with HSP70 itself, that resulted in a high increase in the antitumor immune response after administering a nanovaccine [31]. The authors suggested that the dendritic cells (DCs) take up Hsp70–SPIONs together with tumor peptides and process and load these peptides onto MHC class I and II molecules (Figure 3).

Dextran-stabilized SPIONs targeted with an anti-insulin-like growth-factor binding protein 7 (anti-IGFBP7) single domain antibody were proposed as an MRI contrast for glioblastoma [32]. The proposed system effectively targeted and selectively bound to abnormal vessels within a tumour. An advanced system based on SPIONs, imidazotetrazine alkylating agent (TMZ), and ligands for vascular targets, designed to cross BBB and treat glioma, was also reported very recently [33]. TMZ was entrapped in the PEG–PLGA block copolymer coating of SPIONs. Transferrin or polysorbate-80 were used as vascular targets, allowing the system to cross the BBB. Transferrin expressed on the brain capillary vascular endothelial cells mediates the transfer of iron to the brain, while polysorbate-80 facilitates the nanoparticle’s entrance to the brain through the low density lipoprotein receptor on the BBB. Additionally, anti-nestin antibodies were attached to the system in order to target nanoparticles to the glioblastoma. Transferrin proved to perform better than polysorbate-80. The transferrin and anti-nestin antibody carrying system was shown to have a favorable pharmacokinetic profile and was effective in the reduction of the tumor volume in comparison to the treatment with TMZ.

Another, novel approach was to couple SPIONs with aptamers. Aptamers are poly-nucleotides that act in the same way as antibodies through the hydrogen bonds and van der Waals interactions or due to the so-called hydrophobic effect. In the future, aptamers may replace antibodies in clinical use due to their thermostability, in vitro synthesis, and generally reported higher sensitivity and specificity [34]. To verify this possibility, the effectiveness of both the anti-EGFR (Epidermal Growth Factor Receptor) antibody and aptamer in targeting SPIONSs to the breast cancer cells was studied [35]. The obtained results gave evidence that antibodies are more effective and bind to the cancer cells with a higher affinity. The aptamers-bound SPIONs demonstrated to induce less damage in the normal cells; however, the general level of the observed cytotoxicity of both systems was low. The main problem that the authors reported was the aggregation of SPIONs. As we have pointed out before, that could be prevented by the introduction of a suitable (e.g., PVA) coating.

If antibodies are intended to target membrane protein (e.g., EGFR, as in the example above), then it is highly possible that after an Ab-antigen interaction, the entire complex (including SPION) will be internalized into the cell [36]. This mechanism is known to enhance the theranostic potential of the system.

There are further possibilities to improve the antigen-specific targeting of SPIONs. Some reports [37] showed that attaching scFv (Single Chains Fv) antibody fragments to SPIONs could ameliorate their stability and biocompatibility without any loss in their specificity. scFv contains only the Fab (Fragment Antigen Binding) domain of an antibody, so its length is significantly lower (approx. 5 nm) than that of the whole Ab of IgG class (approx. 28 nm). There is also less chance of inducing an immunological response due to the lack of Fc fragments [38].

We would like to emphasize the fact that proper functionalizing of SPIONs with Ab is crucial for taking advantage of their capabilities. As shown, the antibodies allow us to localize NPs precisely where they are meant to be and thus enable the theranostic approach. Except for cancer therapy, Ab-SPIONs could be also used to purify blood from inflammatory cytokines [39]. 

### 3.2. SPIONs as an MRI Contrast Agent

Magnetic Resonance Imaging (MRI) stands out from various methods of visualizing anatomy (e.g., tumor lesions) in vivo thanks to its lack of invasiveness and good image quality. However, it is still limited by insufficient resolution—some few millimeters-in-diameter changes could be missed with MRI. SPIONs have been already introduced as effective to explain the SPION role in MRI; we need to refer to the basics of this method. 

Certain atomic nuclei, such as ^1^H, possess a property known as spin, which can be represented in a classic approach as a nucleus spinning around its own axis. When introduced into the static magnetic field B_0_, those spins become aligned parallel (a lower energy state) or antiparallel (a higher energy state) to the magnetic field. In room temperature, there are more nuclei in the lower energy state than in the high energy one. As a result, macroscopic magnetization (M_0_) parallel to B_0_ is observed (see Figure 4). When a variable field B_1_ of the frequency which allows to excite the spins is introduced perpendicularly to B_0_, it will cause a transition between the states. As a result, magnetization will be rotated away from the OZ direction to the transverse plane. After the oscillating field B_1_ is switched off, the system slowly returns to the initial state generating FID (Free Induction Decay). Registering the FID signal for protons in water molecules present in the tissue gives 2-D or 3-D information on the spin spatial distribution and is then converted to an MR image using the Fourier transform (FT). 

The stronger the static magnetic field B_0_ used in MRI, the better the signal-to-noise ratio is, as well as the spatial and temporal resolutions of the obtained images. The currently used static magnetic fields are very strong. These used routinely in diagnostic management are typically between 1.5 T and 3 T, but the first 7 T apparatus was recently released for clinical use in Europe and the United States [41]. The 10.5 T magnet was used in the preclinical studies to measure the human brain at the University of Minnesota’s Center for Magnetic Resonance Research in 2017, and even more powerful magnets are still being developed.

In animal studies, a dose as low as 3.3 mg Fe/kg of PVA-stabilized SPIONs was enough to observe a significant improvement in the sample contrast when a 3 T apparatus was used [42], while for a 9.4 T MRI system, the concentrations of SPIONs stabilized with an ionic derivative of chitosan as low as 0.55 mg Fe/kg were sufficient [15]. To use a comparison, Resovist^®^, the FDA (Food and Drug Administration) approved formulation (further FDA issues will be discussed later), is normally administered in the adult patient with a body weight below 60 kg in a dose of 250 mg, which gives an average concentration of 3–4 mg Fe/kg.

For MRI applications, SPIONs are generally coated with poly(ethylene glycol) (PEG), poly(vinyl alcohol) (PVA), or natural polysaccharides (dextran and modified chitosan) due to the long shelf-lives of these systems. For targeting the specific tissue, they could be combined with antibodies or aptamers. There were also trials to spread gold and gadolinium ions [43] on their surface to enhance the contrasting potential. The type of the coating affects the relaxation time. Hydrophilic molecules on the surface stabilize water molecules around SPIONs and lower T_2_. For PEG, the effect increases with chain length to make a particle system a colloidally stable form; there are several molecules that can be used to preserve the integrity and stability of NPs in biological fluids such as PEG, zwitterionic ligands, glycans, aptamers, etc. The strategies to anchor these ligands onto the NPs surfaces are rather diverse and highly dependent on the NP composition, reflecting the different binding affinity towards a specific material. There are recognized three main classes of functional groups (thiols, amines, and hydroxyls) that are used to anchor ligands to the NPs surfaces. In general, thiols are used to covalently bind molecules on the surface of gold and quantum dots. In the case of oxide particles, such as iron oxide, molecules with hydroxyl terminations are used to promote oxygen bonding. On the other hand, many of the employed ligands are not commercially available, requiring very complicated synthetic routes and therefore considerably restricting their applications [44]. Additionally, thanks to the various surface modifications that could be done on SPIONs (e.g., the addition of a fluorescent label), they might become a base for multimodal imaging. Particularly, often they are mentioned together with NIRF (Near Infrared Fluorescence) using the dye Cy5.5 (Cyanine 5.5)-SPION system [45], but also CT (Computed Tomography), PET (Positron Emission Tomography), and SPECT (Single Photon Emission Tomography) are possible to be used in combination with MRI when applying SPIONs as a multimodal contrast [46]

Finally, there is an increasing trend to use SPIONs in diagnostic and clinical techniques. The magnetic technique in which a handheld prototype magnetometer was used to trace SPIONs injected subcutaneously into the breast was shown to be a feasible technique for sentinel lymph node biopsy (SLNB), with an identification rate that was not inferior to the standard technique (radioisotope and blue dye or radioisotope alone). Additionally, the sentinel lymph node dissection (sLND) using a magnetometer and SPIONs as a tracer was successfully applied in prostate cancer (PCa). The feasibility of sentinel lymph node visualization in MRI after intraprostatic SPION injection has been reported. The results of the preoperative MRI identification of SLNs and the outcome of subsequent intraoperative magnetometer-guided sLND following intraprostatic SPION injection were studied in intermediate- and high-risk PCa [47,48,49].

All the abovementioned advantages of SPIONs allow the superparamagnetic iron oxide nanoparticles to be chosen over gadolinium chelates for cell labeling purposes [50]. They are also effectively taken up and do not have a negative effect on the viability or function of the stem cells [51].

Another potent way of using SPION as an MRI agent is to make them the PTT (PhotoThermal Therapy) material [52]. HA (Hialuronic Acid)-SPIONs conjugated with anti-CD44 (Cluster of Differentiation 44) antibodies have been investigated in a breast cancer model [53]. T_2_-weighted MRI image in vivo had about a 40% increased contrast 1.5 h after injection in comparison to the initial state, and the effect lasted for 24 h. Also, the PTT therapy with these systems allowed for the reduction of the tumor volume from above 4000 mm^3^ to an almost undetectable value. The authors emphasize that such effectiveness is partially due to the Ab-antigen-dependent endocytosis of SPIONs.

An even greater improvement in SPION efficiency in affecting MRI contrast was reached as a result of the in vitro research on the LNCaP prostate cancer cell lines. As they are PSMA (Prostate-specific membrane antigen)-positive, these cells could be treated with anti-PSMA Ab-SPIONs (J591 mAb-conjugated SPIONs). The result was a 95% decrease in the intensity of the image in this cell line sample [54]. That result suggests again that developing methods of enhancing the target specificity and avoiding collateral interactions should became the main goal as far as SPIONs are concerned.

There is also an interesting example of combining several new ideas together. SPIONs were not only conjugated with aptamers directed against MUC-1 (marker of colon cancer) but also coated with gold [55]. The main idea of that approach was that gold, as a noble metal, is relatively inert and, thus, should stabilize SPIONs and reduce their cytotoxicity to the normal cells. Indeed, the cell line CHO (MUC1 negative), used as the controls, showed a significantly lower cellular uptake of aptamer-Au-SPIONs than HT-29 (MUC1 positive). Gold coating had an additional advantage of generating heat after excitation by LED (Light-Emitting Diode) and thus allowed for performing PTT. 

In another experiment, SPIONs were coated with folate because the targeted tumor showed a high membranous folate-receptor expression [56]. The results were consistent with the observations described above for other systems. Folate-SPIONs were uptaken into the HeLa (folate receptor positive) cells and significantly decreased the signal intensity providing a negative contrast. The entire effect was dose-dependent.

Besides their oncological use, SPIONs could also facilitate other novel therapeutic approaches which involve MRI. They were, for example, successfully applied in the model of ischemic stroke [57]. Cerebral stroke remains one of the most frequent causes of mortality or severe disability in developed countries [58].

Application of SPIONs as an MRI contrast was also proposed in stem cell therapy, which aims at promoting neurorepair and neuroprotection via the excretion of neurotrophic factors which occurs in the stem cells [59]. As the obliteration of a cerebral artery causes an irreversible damage to the cortex neurons, the proper treatment seems to be replacing them with NSC (Neural Stem Cells). That approach, however, is limited by a low viability of NSC and by the difficulties in following the progress of the treatment [60]. To resolve those problems, siRNA-coated SPIONs were used. siRNA was used for silencing NgR (Nogo-66 receptor), which is responsible for inducing the differentiation of NSC to astrocytes rather than the neuronal cells. The experiment was followed by MRI, and the described results were received partially thanks to the possibility of in vivo imaging.

Imaging of the transplanted stem cells in a noninvasive manner is essential, as it can provide an insight into the cellular proliferation dynamics, biodistribution, migration dynamics, differentiation processes, and participation in tissue repair [61]. SPIONs-based MRI contrast can be applied in order to verify whether the graft implantation has been successful and to track the cells as they migrate to the targeted tissue [62]. The stem cells were labeled with SPIONs using transfection agents such as poly-L-lysine (PLL), Fugene, or protamine sulfate [63]. Other authors, considering that transfection agents adversely affect the cells’ health, used SPIONs with dextran-based polymer coating [64] or citrate-coated superparamagnetic iron oxide nanoparticles in their trials, without the use of transfection agents [50]. 

In a similar experiment, the stem cells were deployed in artificially-triggered ischemic areas mostly in the striatum or in intraventricular or cortical regions. The evaluation of the stem cells application was conducted via both MRI and histological analyses. The idea was to compare the accuracy of MRI in vivo localization of the SPION-labeled stem cells with that achieved with histological methods, such as Prussian blue staining [65]. The results of both tests where coherent, which suggests that SPIONs can be successfully used to evaluate the results of state-of-the-art stem cell therapies in cerebral strokes. 

Other processes, such as the progression of osteoarthritis, could also be followed by MRI with SPIONs-based contrasts. There are also promising reports on the imaging of early-state AD (Alzheimer’s Disease) in vivo [66], which would allow to ultimately distinguish between AD and other diseases manifesting with dementia. 

All the examples mentioned above confirm that SPIONs may significantly extend the possibilities of MRI. Their influence on T2-weighted imaging (T2 is transverse relaxation time ) (and T1-weighted (T1 is longitudinal relaxation time)when in appropriately prepared conditions) and also their ability to increase the general resolution of the image, accompanied by their potential of multidirectional modifications resulting in minimized toxicity and the ability to cross the BBB, suggest that SPIONs might become a new-generation contrast agent for MRI, as well as a fruitful source of inspiration for future inventions in this area.

### 3.3. Magnetic Hyperthermia

Another important way of using SPIONs is magnetic hyperthermia. This method is based on applying AMF (Alternating Magnetic Field). The change of iron’s magnetization against some resistance forces releases heat to the environment (the so-called Brownian and Neel relaxation process). This phenomenon is used in direct cancer therapy or may act as an adjuvant treatment for chemotherapy or radiotherapy. The heat that is released leads to cell damage or results in drug delivery. Superparamagnetics generate more heat than the ferromagnetics under the same conditions because of higher hysteresis losses for one-domain magnetic particles.

There are two general ways to administer SPIONs to induce hyperthermia. The first is injecting NPs directly in the tumor area. The second is to infuse them into the veins, which could provide a more homogenous distribution in the entire circulatory system. 

If SPIONs are targeted to the tumor cells, for example using specific Abs, the procedure will induce a massive accumulation of SPIONs not only around but also within those cells. When SPIONs are set in the magnetic field, thermal energy will be released actually only inside the tumor and in its very close environment. Following this, a collateral damage would affect only a few healthy cells around [64,67]. The whole process raises the temperature inside the tumor to 42–45 °C. It is enough to trigger the death of the cells. Such a high temperature activates numerous pathways that lead to destroying the cells, ensuring the success of the treatment [68]. This is particularly significant due to the fact that cancer cells, containing many somatic mutations, are much more prone to temperature increase (generally if it is above 43 °C) [69].

All the efforts to introduce magnetic hyperthermia was officially introduced to clinical practice and were completed in 2011, when it was officially allowed to be applied in the treatment of glioblastoma (in combination with conventional methods). This method increases the response of the tumor cells to chemo- and radiotherapy, as well as their vulnerability to the attacks of the immune system [70].

In the treatment of recurrent glioblastoma multiform, hyperthermia is used as an accompanying therapy. A so-called magnetic fluid, being basically a suspension of stabilized SPIONs, is injected in the tumor area, and then, AMF is administered. A great advantage is that nanoparticles in the form of stable colloids are introduced only once and only within the tumor. Hyperthermia treatment can be repeated by introducing the patient several times into a magnetic field. With that approach, a 4.5-fold increase in survival was obtained in animal models. The results of the tests carried out on patients also seem very promising, as the significant side effects occurred in only 2 out of 14 patients and even those resolved after 24 h [71]. 

There is an additional effect of magnetic hyperthermia that is worth noting—it was reported to enhance the expression of HSP. As HSP is responsible for higher tumor immunogenicity, it indicates an important additional, positive systemic effect in that local treatment method [69].

Recently it has been shown that cancer tissue is penetrated by the mesenchymal stem cells. This important finding inspired the studies where it was possible to combine the mesenchymal stem cells with SPION preparation Venofer (through endocytosis). Then these cells penetrated into the tumor area, that allowed for performing an ablation using magnetic hyperthermia. When heat killed the stem cells, NPs were released and immediately taken up by the tumor cells, which strengthened the ablation effect. The authors emphasize that the properties of the stem cells were not affected by SPIONs [72].

Magnetic hyperthermia alone and in combination with brachytherapy was used in a clinical setting in the treatment of prostate cancer. Magnetic nanoparticles were administered directly to the tumor. Very promising results have been obtained, but this method is effective only for solid tumors. In the diffuse cases, however, it did not work. A possible solution could be to introduce SPIONs intravenously in combination with antibodies [73]. However, in this study, perineal pain lasting for a few months and rectal fistula were reported. That indicates that SPIONs in high concentration ought to be carefully applied in hyperthermia.

Hyperthermia can also be used to stimulate the anticancer immune response. Referring to the studies that investigated a mouse model of melanoma, raising the temperature of tissue significantly improved the immune response with the participation of T lymphocytes. This therapy was applied before surgical resection and was directed at protecting against recurrences of tumors after removal and at helping to eliminate metastases [74].

Another use of magnetic hyperthermia is to serve as a trigger for drug administration in diseases of the gastrointestinal tract. For example, water-soluble capsules containing a drug were coated with SPIONs in combination with waxy composites. Such coated capsules are resistant to acids, bases, and enzymes in the alimentary canal. To release the drug in a specific place, the radio frequency irradiation is emitted. The RF pulse lasts for a few minutes, that causes the release of heat, dissolution of the wax coating, and dissolution of the capsule’s wall. The melting point of the capsule coating is designed so the heat would not negatively affect the drug molecules before they are released, and it must be, of course, above the body temperature. It could be achieved by mixing with each other eicosane and docosane. After completing this process, SPIONs are excreted safely with the stool. This method has a great potential in the treatment of diseases of the gastrointestinal tract, such as Crohn’s disease, colitis ulcerosa, and cancer [75].

Magnetic hyperthermia can also be used in the therapy of people infected with HIV (human immunodeficiency viruses). This is of a great importance because inexpensive and accessible in all regions of the world, such a therapy can lead to a complete eradication of the virus.

Cytotoxic T lymphocytes (CTL) are important in the fight against the HIV virus. However, with the passage of time, they show the presence of markers of depletion and their ability to kill the infected cells decreases. CTLs specific for HIV act much better at higher temperatures, therefore, magnetic hyperthermia induced by SPIONs can significantly increase the cytotoxic effect, contributing to the decrease in the number of infected cells. However, there was no direct cytotoxic effect on the HIV-infected T lymphocytes following SPION administration [76].

### 3.4. Drug Delivery

Conventional chemotherapy involves a serious damage to the whole organism, so it is important to create targeted therapies, directed selectively toward the cancer cells. The targeting strategy should also increase the effectiveness of therapies by allowing for the administration of higher doses of the drug not limited by the detrimental effect on the healthy tissues. 

SPIONs are undergoing trials in order to prove that they could be implemented as drug carriers. In order to deliver drugs to the selected location, an external magnetic field can be used, which allows for concentrating SPIONs at the selected location. The properties of SPIONs and the success in delivering drugs are strongly affected by the composition of the external coating, regardless of whether it is formed by a thin polymeric layer, multilayered film, or more sophisticated carrier system (capsules, particles, and vesicles). It is advantageous to use biocompatible polymers, either natural or synthetic, as they help to protect nanoparticles against being recognized by the immune system. They are also responsible for encapsulating drugs, as well as for their release under appropriate environmental conditions or in response to external stimuli [77].

There are several important criteria that have to be considered in order for the SPION-based drug delivery system to be effective. The coating/carrier should provide the delivery system with a suitable hydrophilicity, so it can be easily dispersed in aqueous media. It should also provide functional groups which can be further modified in order to control the drug release or to bind targeting units [78].

One of the solutions which meet the abovementioned requirements is to use thermally crosslinked SPIONs (TCL-SPIONs). In this system, a copolymer having thermally crosslinkable Si–OH groups in its structure (poly(3-(trimethoxysilyl)propyl methacrylate-r-PEG methyl ether methacrylate-r-N-acryloxysuccinimide)) was used as a stabilizing coating for SPIONs. This approach proved to be a good method to ensure a suitable dispersion of SPIONs in water and to introduce the functional carboxyl groups to the surface of the magnetic nanoparticles [79]. A new idea was also proposed to investigate an interaction between β-cyclodextrin-conjugated SPIONs (CD-SPION) and polymerized paclitaxel (pPTX). It resulted in forming a kind of nano-assembly, which acted as a protective barrier for the entire structure inside the vessels. Thanks to this combination, an effective magnetically-induced targeting effect was obtained and an increased antitumor activity was achieved both in vitro and in vivo. The high magnetization obtained made it possible to use this system for deeply localized tumors [80].

Another very interesting proposition are SPIONs with a polymeric shell made of PEG and PEI (poly(ethylenimine)), which was modified with folic acid (FA) using the EDC (1-ethyl-3-(3-dimethylaminopropyl)-carbodiimide)/NHS (N-hydroxysuccinimide) chemistry (Figure 5). This method is very likely to be successful in the treatment of tumors overexpressing the FA receptors. Folic acid, by combining with its receptors on the surface of the cells, facilitates endocytosis of the systems, which allows for the delivery of drugs directly to the cancer cells. The results of the tests carried out on cell lines were extremely promising—the absorption of drugs by the cells increased significantly. Unfortunately, in vivo tests revealed some problems. The use of this method involves placing the magnet in the tumor focus, which turns out to be difficult for deeper-situated tumors. In their case, usually too few SPIONs reach the tumor area. Animal studies have already proved that this method worked well for breast cancer in mice [81].

One of the most common cancers in men is prostate cancer. The treatment modality based on the effects of androgens on its development eventually leads to conversion into castrate-resistant prostate cancer (CRPC). Then, the treatment is based on the administration of docetaxel (Dtxl), which is unfortunately very harmful to the entire organism. Another serious problem results from the fact that CRPC cancers are often resistant to this drug. It has been proved that targeted therapy using SPIONs allows for delivering Dtxl to the cancer cells, saving other tissues and additionally increasing the sensitivity of the tumor to the administered drug. SPIONs are conjugated with docetaxel, as well as with antibodies against the PSMA antigen present on the surface of the tumor cells. SPION-Dtxl caused a significant increase in proapoptotic proteins, while reducing the level of antiapoptotic ones in the prostate cancer cell lines. A significant difference in drug activity was also evidenced by the comparison of the PSMA^-^ - (PC-3) and PSMA^+^ (C4-2) line cytotoxicity—the drug is much more efficient against the lines containing this antigen. A similar result was obtained in ex vivo studies, which confirms the great potential of this targeted therapy [82].

Other diseases in which SPION therapy may revolutionize the treatment schemes are arthritis and osteoarthritis. The use of SPIONs could allow for a better control of the concentration of drugs in the joint cavity and may prolong the presence of the drug at this site. One of the systems proposed are poly(lactic-co-glycolic acid) (PLGA) microparticles containing co-encapsulated dexamethasone acetate and SPIONs. One of the possibilities of maintaining such magnetic particles in the joint cavity is an external magnet, also controlling the release of the drugs [83].

Cisplatin, a popular chemotherapeutic agent used in the treatment of solid tumors, unfortunately has a damaging effect not only on the cancer cells but also on the healthy tissues. A big problem is also its low solubility in water and low lipophilicity, as well as the fact that the cancer cells may show resistance to its action. In order to release cisplatin in the cancer cells, a new version of magnetic nanoparticles was created. Porous hollow nanoparticles (PHNP) were formed as a result of the controlled oxidation of iron in nanoparticles followed by acid digestion. After placing the drugs inside them, these structures were stable under physiological conditions, considerably slowing down the drug. However, at low pH, the pores opened and the content was released much faster. In the human cells, such pH values could be found in the lysosomes. 

The outer surface of the nanoparticles was used to direct SPIONs to the tumor focus (these methods depend on the type of cancer) after the cells absorbed the nanoparticles, and then, after the lysosome was formed, cisplatin was quickly released, which significantly increased its cytotoxic effect. Another advantage of this cisplatin protection is the fact that the drug is not inactivated by the plasma proteins before it reaches the tumor. This method has shown promising results in the treatment of breast cancer [72].

### 3.5. Other Usages of SPIONs

To provide a thorough insight into the issue of SPIONs’ medical application, the literature reports on the new ideas of how these nanoparticles can be used are presented in the following paragraphs.

#### 3.5.1. Alzheimer’s Disease Therapy

Although the capability of SPIONs to penetrate into to the brain might entail some risks, it sets them among promising agents against brain tissue diseases, for example those involving the accumulation of peptide amyloids. 

In an innovative study, Italian and Slovakian scientists discovered that magnetic Fe_3_O_4_ nanoparticles are able to interact with lysozyme amyloids in vitro leading to a reduction of the amyloid aggregates, thus promoting depolymerization. The studied nanoparticles also inhibited lysozyme amyloid aggregation. The probable mechanism of the anti-aggregating action of these NPs has been proposed. Lysozyme molecules are adsorbed and adhere to the nanoparticles. Consequently, a significant decrease in a free lysozyme molecules’ concentration can be observed, which hampers the nucleation process and fibrillogenesis [84]. SPIONs are one of the few NPs which do not trigger a reverse effect (e.g., increased amyloid aggregation after supplying TiO_2_). Undoubtedly, many secrets await the scientists in this field, and more experiments are needed to be done to explore the detailed molecular mechanisms of the anti-amyloidogenic ability of the NPs.

#### 3.5.2. Photodynamic Therapy

As mentioned in previous paragraphs, SPIONs can serve as efficient drug carriers. Interestingly, these nanoparticles can also deliver photosensitizers in photodynamic therapy.

In a deft combination of magnetic drug targeting (MDT) and photodynamic therapy (PDT), SPIONs were covered with dextran coatings to prevent them from aggregation and to allow for the linkage of hypericine (photosensitizer) to its hydroxyl groups. In this approach, as the authors suggest, a significant selectivity and fewer side effects of PDT should be observed, as the therapeutic effect occurs only in the area where the particles accumulated by the external magnet are under the influence of certain radiation produced by a laser system [85]. 

The in vitro toxicity of free and particle-bound hypericin, as well as SPIONs themselves, was investigated in the nonadherent human T-cell leukemia cell line Jurkat. The amount of light-induced, hypericine-produced ROS was determined using the fluorescence dye DCFH-DA (Dichloro-dihydro-fluorescein diacetate assay).

Although the cytotoxicity of SPION-bound hypericin proved lower than in the controls (with hypericin only), the authors suggest that longer irradiation would change it and augment the therapeutic effect. 

#### 3.5.3. Cytotoxicity in Osteosarcoma Cells

Although SPIONs are often referred to as nontoxic nanoparticles, an interesting study showed that the overendocytosis of SPIONs induced death through various mechanisms in the osteosarcoma cells during the application of a spinning magnetic field [86]. These effects, once considered as adverse, can be turned into a possibly effective therapy. The cause–effect relationship between SPIONs and autophagy or apoptosis was proved using 3-methyladenine as the autophagy inhibitor and Ac-DEVD-CMK (Ac-Asp-Glu-Val-Asp-chloromethylketone) - cell-permeable, and irreversible inhibitor of caspase used to block apoptosis. These substances increased the viability of the cells exposed to SPIONs and a spinning magnetic field. 

#### 3.5.4. Delivering Linoleic Acid in Breast Cancer Therapy

Superparamagnetic iron oxide nanoparticles (SPIONs) have been described as useful for diagnostic assays, local hyperthermia-based cancer therapy, and the intra-tumor delivery of conventional drugs or antibodies. Another idea emerges from a study of the breast cancer cells in which the cytostatic effect occurred when SPIONs with conjugated linoleic acid (CLA) were administered. It was correlated with the increase of peroxisome proliferator-activated receptors γ (PPARγ) as CLA is one of their natural ligands. Some other changes, such as the downregulation of p-glycoprotein (it is the so-called MDR—Multi Drug Resistance protein, which increases the drugs’ efflux out of the cells) [85,87] and drug metabolizing enzyme ALDH3A1 ensued, which opens up new possibilities in the management of breast cancer drug-resistance issue [88].

#### 3.5.5. SPIONs Interfering with Electron Transport Chain in Hepatic Carcinoma

Another study shows that SPIONs can preferentially interfere with the mitochondrial electron transport chain (METC) in hepatic cancer cells. Several major changes in mitochondrial functions have been reported. SPIONs impaired ATP generation and decreased the MMP (Matrix Metaloproteinases) level. Abnormal perinuclear clustering of Ca^2+^ and cytochrome c were observed. Interestingly, the SPION treatment did not alter the activation of caspase-3 in the QSG-7701 normal cells but increased Bax and cleaved caspase-3 in the HepG2 cancer cells, promoting their apoptosis. Furthermore, SPION treatment significantly depleted the cancer cell GSH levels but did not affect the QSG-7701 cell level of this major anti-oxidative factor [89]. Although these results are promising, further research ought to be conducted because many mechanisms of SPION-induced, cancer-specific cytotoxicity still remain unclear.

#### 3.5.6. Remote Control in SPIONs

Remote control of nanoparticles’ movements might have far-reaching implications in numerous areas of nanotechnology and can prove to be another way of inducing apoptosis of the tumor cells. It has been reported that SPIONs covalently conjugated with antibodies targeting the lysosomal protein marker LAMP1 (LAMP1-SPIONs) rotate in a dynamic magnetic field and cause lysosomal membrane damage in the rat insulinoma tumor cells. This disturbance increased the expression of early and late apoptotic markers and reduced the cell growth [90]. 

#### 3.5.7. Detection of Aflatoxin 

Antibodies-coated SPIONs have been used in numerous studies. In one of them, fluorophore labeled anti-aflatoxin antibodies-coated SPIONs have been synthesized for the detection of aflatoxin B1. Aflatoxin B1 is one of the deadliest mycotoxins known to be present in groundnuts, corn, animal feeds, and milk of lactating animals contaminated with *Aspergillus flavus* and is classified as a group 1 carcinogen by The International Agency for Research on Cancer (IARC) [91]. The authors have attempted to detect the aflatoxins at very low concentrations, at a nanoscale level. This approach might be successful also in the detection of other antigens. 

#### 3.5.8. Prevention of Bleeding after Application of Heparin-Based Drugs in Hemodialysis

Extracorporeal circuit, the treatment of thromboembolic events, and many other procedures require heparin-based anticoagulant drugs for the prevention of blood clotting. Unfortunately, during hemodialysis, these anticoagulants migrate to the patient’s blood and elevate the risk of bleedings. SPIONs can be used to create anticoagulants for hemodialysis, which are removed before transfusing the blood back to the patient’s body. In an attempt to create such a factor, Fe_3_O_4_ nanoparticles were synthesized in a solvothermal reaction in the presence of positively charged PEI, followed by a self-assembly with negatively charged heparin. As a result, the heparin-SPIONs (Hep-SPIONs) consisted of a heparin polymeric shell encapsulating multiple SPIONs inside. These nanomaterials were designed to be magnetically directed to the desired sites by an external magnetic field. As a result, they can be effectively removed before flowing into the human body, and the risk of bleeding can be distinctly reduced [92].

#### 3.5.9. SPIONs against Bacterial Diseases

SPIONs are listed among nanoparticles with antibacterial properties. For instance, gold-coated SPIONs exert strong toxic effects on bacterial biofilms by penetration into these structures. Both SPIONs’ cores and the intermediary gold shell have the capability to induce heat when an alternative magnetic and laser fields are applied. The evoked high temperature can be used as an additional factor, which may intensify the deadly effect on bacteria caused by these NPs.

The antibiotic resistance considerably challenges the treatment of numerous infections and poses a threat to global health, food security, and development. Therefore, steps are taken to facilitate the rapid and thorough identification of bacteria so that specific, targeted treatment can be introduced. The magnetic labeling of bacteria in suspension allows for their effective extraction from water solutions. It is conducted with the use of an appropriate magnetic field, such that it makes it possible to remove even low numbers of bacteria [93]. 

In some studies, SPIONs with various functionalities, such as carboxyl and amine groups, were used [94]. Amine functionalized SPIONs achieved over 97% efficacy in bacteria capture. One of the most effective methods described involved the use of cationic magnetic nanoparticles (amine functionalized SPIONs) and commercially available magnetic cell separation (MACS) columns. The cells that had been labeled with superparamagnetic nanoparticles (SPIONs) were immobilized when they were passed through the column in aqueous suspension. The positive charge of SPIONs’ shells ensured their efficient binding to anionic domains on the bacterial cell’s surface. 

Apart from such methods that allow for the efficient but non-specific capture of bacteria, some other applications have also been proposed. Especially promising are those with the use of SPIONs covered with antibodies, such that specific pathogens, e.g., *E.coli* O157:H7, can be labeled.

#### 3.5.10. Magnetic Particle Imaging

While in MRI, SPIONs are used merely to improve the contrast of the obtained images; in magnetic particle imaging (MPI), they are the only source of signal and the only visualized element. In 2005, Gleich and Weizenecker reported creating the first MPI images and successfully managed to prove that this imaging technique was feasible [95]. The SPIONs in MPI are referred to and act as tracers rather than contrast agents. In a 1.5 T field, SPIONs have an approximately 10^8^ times higher magnetization and 10^4^ times faster relaxation than protons, which are the main source of an MRI signal. As a result, an excellent temporal resolution and a higher signal-to-noise ratio (SNR) are achieved. Another vital feature of this method is that MPI visualizes only the anatomical structures, which are labeled by the tracer [96]. MPI is sensitive, works without the use of potentially harmful ionizing radiation, and offers a three-dimensional image of the SPIONs’ distribution with a fine contrast. These characteristics determine MPI as useful for a wide variety of medical applications. MPI offers a promising alternative to conventional methods in cardiovascular diagnostic and interventional procedures, as well as opens up new possibilities in cell labelling and tracking. One of the first SPION preparations, which showed an acceptable MPI performance, was Resovist (Bayer Pharma AG). Since then, however, many other SPIONs with a performance better than Resovist have been synthesized [97]. In contrast to MPI, CT (Computed Tomography) and MRI images provide anatomical information. The fusion of MRI or CT and MPI can be interesting and may prove beneficial, especially in perfusion-studies and cell tracking. MPI could provide quantitative and sensitive tracer images, if needed, with a high frame rate, while MRI/CT would provide anatomical information with superb tissue contrast [98,99,100].

## 4. Side Effects of SPIONs

Iron oxides can be found in the form of nano-sized crystals in the earth’s crust. They are generated in a variety of natural processes, so it would seem that the use of NPs containing iron oxides should carry no serious risk [20]. However, as Paracelsus stated, *dosis facit venenum*. The exposure rate of iron oxides during SPION treatment may be much higher than the one occurring on an everyday basis. More and more NPs are being manufactured to meet the increasing demands of the rapidly advancing nanomedicine. All the therapeutic benefits that SPIONs undoubtedly have to offer go hand-in-hand with the concerns associated with the patient’s exposure to them during experimental therapies [101].

Although from 1996, when the first SPION-derived MRI contrast was approved by the FDA, a number of other products appeared on the market (e.g., Resovist, Lumirem, and Feridex); most of them were discontinued due to safety issues. The concerns emerged about the impaired mitochondrial function, the appearance of apoptotic bodies, the generation of ROS, and DNA damage [17]. Currently, the only FDA-approved product is ferumoxytol, which is used to treat iron-deficiency anemia in patients with chronic kidney disease and off-labelled as an MRI, angiography agent in patients with renal failure [102]. 

Therefore, there is a considerable need to address the biocompatibility and safety issues associated with the use of SPIONs and to name several side effects typical of therapies involving the use of SPIONs. 

Iron oxide nanoparticles are usually taken up by macrophages in the mononuclear phagocytic system (MPS) of the liver, spleen, lymphatics, and bone marrow. The blood half-lives of SPIONs vary from 1 h to 24–36 h and are dependent on both the surface coating material and the size of nanoparticles. Smaller NPs have longer half-lives than larger ones with the same coating [103]. Different mechanisms of SPIONs’ uptake have been proposed: passive diffusion, receptor-mediated endocytosis, clathrin-mediated endocytosis, caveolin-mediated internalization, and other types of the clathrin- and caveolin-independent endocytoses [104]. In the studies concerning SPIONs’ internalization, various endocytic inhibitors were analyzed. For example, chlorpromazyne, as an inhibitor of clathrin-dependent endocytosis, significantly reduced the uptake of SPIONs [105]. Once taken up by the MPS cells, the iron oxide nanoparticles are degraded in the lyzosomes. Dextran coatings, for example, are processed by intracellular dextranases, and their remnants are excreted in urine. In the same time, the iron is incorporated into the body’s iron storage and can be found, for example, in the red blood cells in hemoglobin [106].

Hong et al. [107] have studied a series of SPIONs stabilized with various coatings; the nanoparticles differed also in surface charge and size. The performed in vitro studies allowed for concluding that at concentrations below 200 μg/mL, all the tested SPIONs exhibited no cytotoxicity and that up to 500 μg/mL, their use in biological applications was considered relatively safe. It was also noted that nanoparticle concentration was more critical than other factors, such as surface modification and size. The comprehensive cytotoxicity review studies on SPIONs by Patil et al. [108] showed that SPIONs exhibited no toxicity at concentrations below 100 μg/mL, while in some cases the concentration of 250 μg/mL considerably decreased cell viability. The dependence of toxicity on the nanoparticles’ coating was also evidenced. 

One of the most important toxic effects induced by iron oxide nanoparticles is their ability to cause oxidative stress. The free iron in the form of ferrous ions (Fe^2+^) can react with hydrogen peroxide and oxygen produced by the mitochondria and create highly reactive hydroxyl radicals and ferric ions (Fe^3+^) via the Fenton reaction. Hydroxyl radicals generated by the free iron could damage DNA, proteins, lipids, and polysaccharides in the cellular structure [109]. However, there are not many studies concerning the DNA damage and oxidative stress caused by nanoparticles [110].

Surprisingly, researchers found little correlation between in vivo and in vitro toxicity results in some studies. For example, in rats, the results of in vivo pulmonary toxicity studies demonstrated that the instilled carbonyl iron (CI) particles produced little toxicity and caused only mild inflammatory responses, whereas the results of in vitro pulmonary cytotoxicity studies showed a variety of responses in different cell lines, visible especially at high doses [111]. Probably during in vivo assays, the organism’s ability to maintain homeostasis (e.g., by storing the excess of iron) usually reduces the observable negative effects of SPIONs. According to Mahmoudi et al. [112], the poor correlation between the toxicity measurement methods for in vivo and in vitro experiments may also be caused by the nanoparticles’ ability to change the composition of cell mediums during in vitro assays as a result of the serum proteins binding to the negatively charged uncoated SPIONs. In contrast to studies concerning cell lines, during in vivo assays, changes in the contiguous tissues are less visible, even when identical amounts of nanoparticles are applied. The protein absorption on the surface of SPIONs can distinctly distort the results of in vitro cytotoxicity studies. In an attempt to achieve more reliable and exact cytotoxicity assessment results, a modified method has been reported [113] in which the iron oxide NPs were introduced to the cell medium and the solution was kept in contact for 24 h in order to create a stable protein layer on the surface of the SPIONs. In the next step, the medium was replaced with a fresh one, and only then were the obtained SPIONs employed for toxicity assays. The researchers took into consideration that NPs can cause significant changes in the cell medium, such as the denaturation of proteins, which can result in cytotoxicity. When this modified approach was used, the toxicity of NPs was found to decrease significantly. 

The number of in vivo studies of SPIONs’ toxicity performed in humans is very limited. Ferumoxtran-10 is a commercial dextran-coated reticuloendothelial system-targeted MRI contrast agent consisting of ultrasmall superparamagnetic iron oxide nanoparticles (USPIO nanoparticles) especially developed for MR lymphography [114]. One investigation found that Ferumoxtran-10 induced adverse effects in patients, namely headache, back pain, vasodilatation, urticaria, diarrhea, and nausea, all of which were rather mild, short in duration, and relatively rare (experienced by 6% of patients). As expected, the mean values for serum iron, total iron binding capacity, and percentage saturation increased significantly from the baseline in the 24–36 h following contrast agent administration [115]. 

Targeting of SPIONs to particular areas results in a locally high concentration of iron. Consequently, this significant overload of iron, especially in the form of free ions, can have toxic implications in the exposed tissue and may result in the disruption of iron homeostasis, causing pathological cellular responses. The possible adverse effects include cytotoxicity, oxidative stress, epigenetic alterations, and inflammatory reactions. Even when cytotoxicity is not observed, the exposure to iron ions can still lead to subtle but harmful cellular changes in the form of DNA damage. Another possible source of toxicity is the surface coating of SPIONs. The reports suggesting that anti-dextran antibodies are present in individuals after USPIO administration imply that indeed, there is such a risk. Nevertheless, although dextran remains one of the most frequently used coating materials of the SPIONs used in clinical experiments, no anaphylactoid reactions have been reported so far [116].

## 5. Conclusions and Future Prospects

The role which SPIONs are supposed to play in the future clinical practice is still unclear. Firstly, they could not be generalized, as we have already proved that their particular properties and use depend on the structure of the core (1), coating (2), and attached agents (3). Each of them affects the parameters of the others. 

As very few SPIONs have been introduced to the clinical practice so far, there emerges a question what the actual reason for that state is. Generally, the toxicity of SPIONs is still not clearly described and, for example, for Ferucarbotran, a slight increase in stem cell viability and proliferation was even reported [115]. What is a bit confusing, the authors concluded, is that it is not surely a positive result (due to the theoretically higher tendency to be tumorigenic). 

Nevertheless, while considering the antitumor therapy, the presence of side effects seems almost inevitable. Just an example of one of the most popular anticancer drugs, doxorubicin, supports that claim. Damage done to the brain, liver, alimentary canal, or skin is significant, and their symptoms (emesis, loss of hair, or neurological impairment) have been observed [116], even though a need to overcome cancer justifies applying doxorubicin. As we discussed before, adverse effects of using SPIONs are present, but we did not find them severe or frequent.

The frequently stressed limitations of SPIONs—their uptake by reticuloendothelial system—could be avoided by a proper assortment of coatings. The results of research on chitosan-coated SPIONs suggest that the possible solution has been already found.

Given the circumstances, it seems that the main restriction for introducing SPIONs is that they rely in most of cases on the specificity of some targeting moieties (Abs, aptamers, receptors, and their ligands). Until they are not present in the place where needed, they seem to lose a majority of potential applications.

Therefore, we suggest that there remain two great challenges which have to be responded to before the full potential of SPIONs could be further explored (focusing on oncological practice): 1. The efforts to find some tumor-specific antigens have to be increased. They are the absolute basics for applying SPIONs in MRI, drug delivery, magnetic hyperthermia, and most of the other applications of SPIONs. We could give some examples of antigens that could be targeted in such a way: CD44 (1) (surface cell protein which have 10 different juxtamembranous isoforms—some of them are reported to be present only on specific cancer cells) [14], PSMA (2) (Prostate Specific Membrane Antigen) [117], N-cadherin (3) [118], mesothelin (4), or mHSP70 (5). This approach should be supported with the prevention of phagocytosis by macrophages. 2. The possible combinations of particular components in three-ply (core, coat, and attached molecules) SPIONs have to undergo more research that will compare (even if in vitro) the properties of different combinations of the abovementioned components. The comparison of the already reported experiments could not give us the required information—either due to the different methods or too narrow a range of evidenced combinations. Introducing those ideas would allow for considerable broadening of the clinical use of SPIONs and open up new possibilities not only for administering SPIONs but also for the entire nanotechnology industry.

## Figures and Tables

**Figure 1 materials-12-00617-f001:**
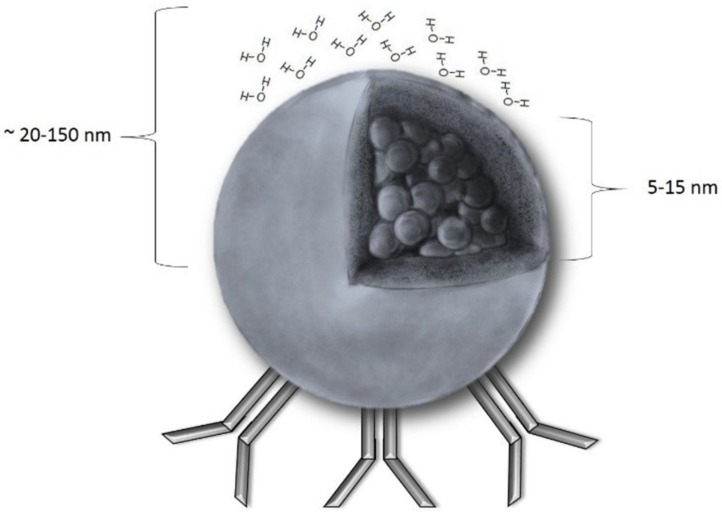
A schematic presentation of a SPION (Superparamagnetic Iron Oxide Nanoparticle): The core radius ranges from 5 to 15 nm, and the hydrodynamic radius (core with shell and water coat) is between 20–150 nm. Unless there is a magnetic field, magnetization equals 0. As shown, SPIONs could be easily coupled with antibodies that facilitates the majority of the SPION applications discovered so far.

**Figure 2 materials-12-00617-f002:**
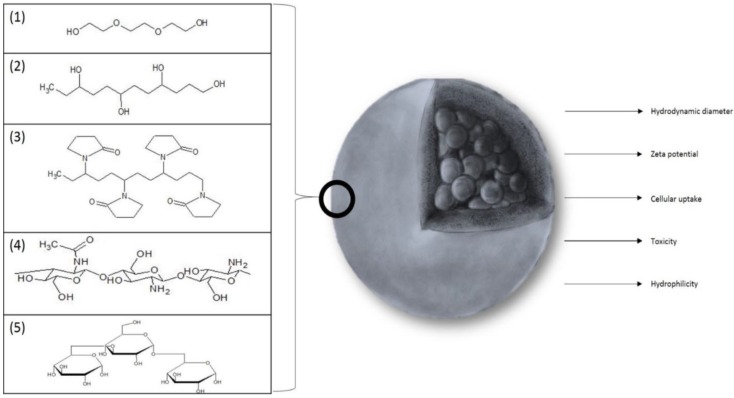
The substances most frequently used as a coating for SPIONs: (1) PEG (Poly(Ethylene Glycol)), (2) PVA (Poly(Vinyl Alcohol)), (3) PVP (Poly(Vinyl Pyrrolidine)), (4) chitosan, and (5) dextran. The figure also sums up the most important parameters that are affected by coating.

**Figure 3 materials-12-00617-f003:**
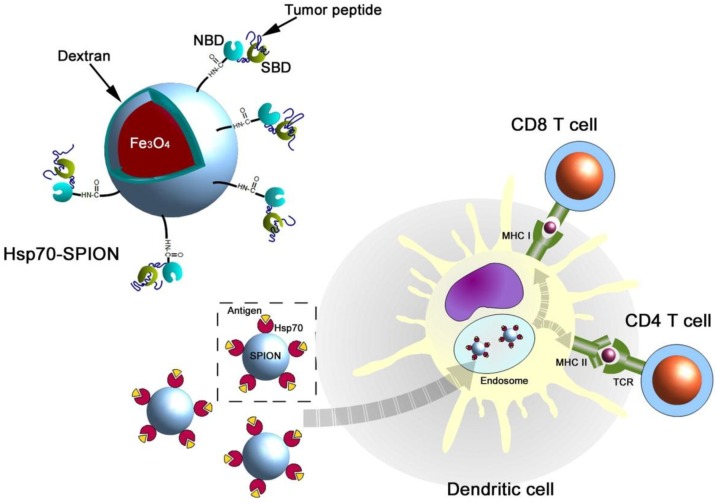
A schematic representation of the antigenic peptide delivery by Hsp70–SPIONs into the dendritic cells (reprinted from Reference [29] with permission from Elsevier).

**Figure 4 materials-12-00617-f004:**
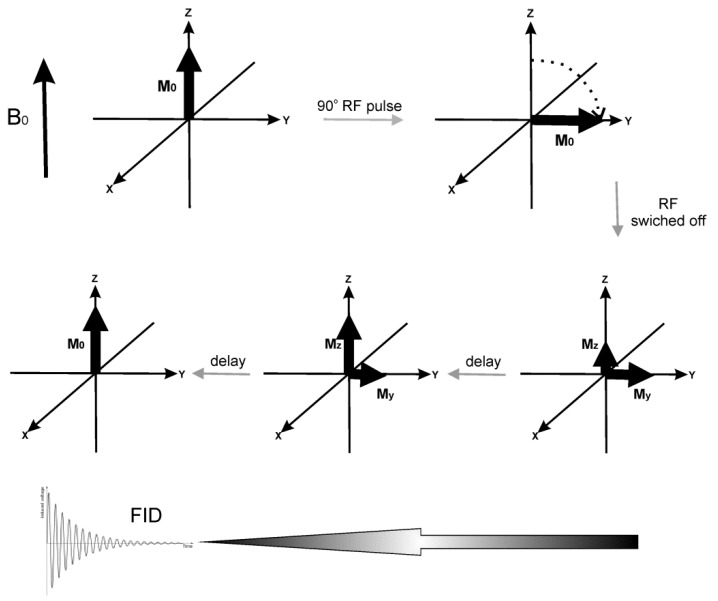
The scheme presenting the main idea of MRI. Upper part: The variable B_1_ field in the form of a radiofrequency (RF) pulse rotates the magnetization to the transverse plane. Lower part: After the RF pulse is switched off, the system relaxes in time until equilibrium is reached, generating a FID signal. The processes leading to thermal equilibrium after RF pulse can be described as longitudinal (spin-lattice) relaxation, with characteristic constant T_1_, and transverse (spin-spin) relaxation characterized by T_2_. The first process occurs due to the dissipation of the absorbed energy to the surrounding environment, while the second results from the energy redistribution among the nuclei. The MRI contrasts improve the quality of the MRI image via influencing T_1_ or T_2_ of the surrounding tissue. Water protons in different tissues have different T_1_ and T_2_, a phenomenon that provides MRI contrast. T_1_ is shortened by the interactions of the observed nuclei with paramagnetic agents and by a limited mobility of H_2_O molecules (observed in more viscous tissues). T_2_ is also shortened by lower molecular mobility. SPIONs are mainly negative T_2_ contrast agents. When accumulated in the tissue, they shorten T_2_, decreasing signal intensity, and provide a negative contrast enhancement. It is also suggested that if SPIONs’ particles are small enough (<10 nm), they could act as T_1_ positive contrast agents [40].

**Figure 5 materials-12-00617-f005:**
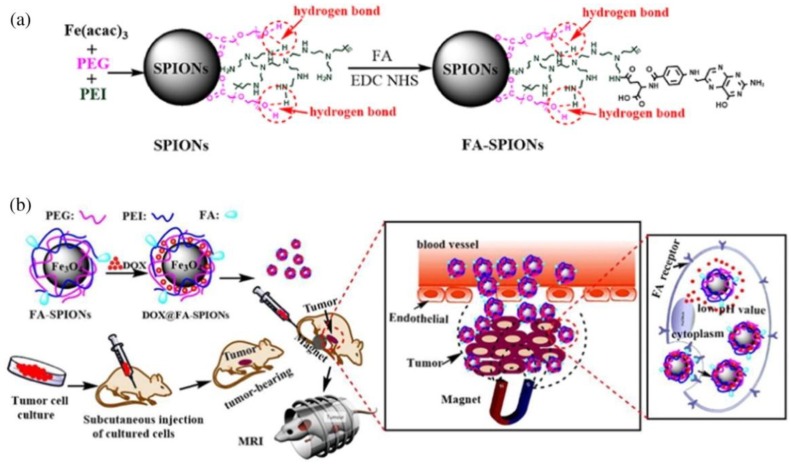
(**a**) The synthesis and surface coating of SPIONs and FA-SPIONs; (**b**) The DOX-loaded FA-SPIONs (DOX- anticancer drug, doxorubicin) for FA-mediated and magnetically-targeted drug delivery to the tumor and the MR imaging (reprinted from Reference [73] with permission from Elsevier).

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
