# Peer review of "Superparamagnetic Iron Oxide Nanoparticles—Current and Prospective Medical Applications"

_materials, 2019, doi:10.3390/ma12040617_

Round 1

Reviewer 1 Report

The review describes selected properties of SPIONs, their bioconjugation and medical applications. This is a reasonable overview but it needs a lot of corrections.  

Major comments:

1a. English could be improved, structure of many sentences is awkward hence difficult to follow. Examples: “SPIONs lose quickly their magnetization after turning down the applied magnetic field, what allows for a precise remote control over their action

“Ferromagnetic materials have aligned magnetic moments if there is no external magnetic field (its magnetization moment M shows a hysteresis loop in function of external magnetic field H).” “There is a huge variety of small compounds that are possible to be attached to iron oxide core. “ These are just examples of improper grammar (sentence structure).

 1b: Statement: “We strongly believe” (l. 325 and at the end) is not acceptable in a scientific paper. Conclusions must be based on scientific facts.

1c. Adding quantification of toxicity (eg concentration, size), detection limits by MRI (what concentration of SPIONs could be detected by MRI and at what field) etc.. would add value to ms.

 1d. I suggest the review is corrected and improved by a senior researcher.

Minor comments:

2. l. 38 – provide references to: “… to bind anti-tumor antibodies to SPIONs surface and inject the resulting, targeted particles into the circulatory system”

 3. l. 50: When used as a contrast agent in MRI, the signal received from SPIONs – MRI does not detect signal from SPIONs, it detects signal from surrounding protons. A method proposed by Gleich, 2005 is not an MRI method.  

 4. l.52: “Velocity of bloodstream, as well as vessels diameter, are another restrictions.” - why “they” are “restrictions”?

l. 53: “In the last few years the understanding of the mentioned above phenomena increased significantly, improving clinical efficiency of SPIONs” – the above “phenomena” is based on incorrect assumptions (“signal received from SPIONs”) hence the incorrect statement. 

 5. Chapter: “Physicochemical properties of SPIONs”

Could the authors describe in more details how coating affects the parameters listed in fig. 2 (diameter, toxicity, zeta potential, cellular uptake etc)? This would increase value of the review.

 6. l. 51: correct the sentence:  “Our goal is to compare results of different researches a in particular field of action,..”

 7. section 3.1 – provide more examples of SPIONs applications to diagnosis of other cancers, such brain cancers (eg glioma), note: it is possible to overcome issue of BBB crossing using vascular targets – provide references.

 8. l. 232: “Then a variable field (1) is introduced, perpendicular to B0 (parallel to OY axis), which is characterized by the induction B1” – the B1 is the actual variable (RF) field. “induction: in not the correct word; please correct the sentence.

9. l.233: “This oscillating magnetic field changes the direction of the magnetization flipping it to the transverse plane” – This field flips the direction. But not “changes flipping” – please correct..

 10. l. 238: correct both grammar and context: “The currently used (with SPIONs) static magnetic fields are very strong, the most frequently noted value being 9,4 T. “ … this refers to preclinical MRI only!

General comment: the section on MRI could be improved. It is confusing now. Points 8-10 are examples only

 11. l. 328: lack of toxicity, - this is not true, it depends on concentration; ability to cross BBB - also not true, it depends on many factors. Toxicity is mentioned later in the text, yet the sentence is incorrect.

 12. Please discuss recent FDA issues with SPIONs approvals.

 13: Fig 4 – why My=M1?

Author Response

We would like to thank the Reviewer for reviewing our manuscript and for pointing out  

to the issues  needing clarification or further explanation. We believe it will help us to improve our manuscript. All responses to the reviewer's comments in the attachment

Reviewer 2 Report

With delight I reviewed the manuscript by Dulinska-Litewka et al. describing the elementary properties of SPIONs and the current role in medical diagnostic and treatment. The present review provides a comprehensive overview of current experimental and clinical use of SPIONs and future perspectives. Also cons or possible limitations are discussed in detail.

In general, the manuscript is concise, well written and technically sound.

However, some minor issues should be taken into account to improve the value of this work.

Minor comments:

- 3. Summery of clinical applications of SPIONs:

The authors provide an extensive lineup of clinical applications of SPIONs. However, for the Reader, it is not always easy to distinguish between experimental approaches and already established procedures. In principle, a corresponding clear structure would be very helpful.

Furthermore, the clinical use of SPIONs for visualization and magnetometer-guided intraoperative detection of sentinel nodes has not yet been mentioned by the authors. SPIONs are already used for this purpose in different tumor entities and various research activities are on the way (e.g., Pouw et al., Int J Nanomedicine 2015; Douek et al., Ann Surg Oncol 2014; Pouw et al., Br J Radiol. 2015; Cousins et al., Sci Rep 2015.; Winter et al. Int J Nanomedicine, 2018; Winter et al., Molecules 2017; Winter et al. Eur Urol 2018). Therefore, this issue should be supplemented in the paragraphs on therapy and magnetic imaging/MRI.

- The prostate-specific membrane antigen should be uniformly named and abbreviated (“PSMA”).

- Some text passages were underlined. The underlines has to be removed.

Author Response

(The authors gave the same response as above.)

Reviewer 3 Report

Dear Authors,

the presented manuscript reviews current different applications of SPIONs. Unfortunately, there are numerous typos and style errors (double spaces, different font sizes, underline of text, etc.) throughout the whole manuscript, indicating that the manuscript was not carefully reviewed by the authors themselves before submitting the article. This for itself can be a reason to reject the paper from being reviewed. Nevertheless I have a couple of more questions and comments on the manuscript:

General: Personally I don´t prefer the structure of the manuscript, which is in part quite confusing and in part it misses the discussion. For example, a lot of applications of antibody-related SPION applications are presented, but the downside of side effects and costs are not mentioned in the manuscript. The section 3.5 is not really necessary in the presented form, since most of its subsection can be included in the other parts.

l. 78ff: An image showing the different magnetizations as a function of external magnetic field would be nice.

l.83ff: Hematite possesses only low magnetic properties, which is why it is scarcely used in application related to its magnetic properties

l.88f: The potential at the surface of the particle is referred to as stern potential. The zeta potential is the potential at the surface of hydrodynamic shear and can be used to quantify surface charge.

l.90f: What about the increased toxicity of positively charged particles?

l.125f/l.312/f: There are also highly biocompatible dextran-coated SPION formulation that are highly biocompatible, even at concentrations higher than 400 µg/mL. This means it strongly depends on the dextran-derivative and surface binding.

l.138f: There are also SPION formulations that are coated with human serum albumin. So if want to discuss protein adsorption in your Review you should at least mention them.

l.174ff: Missing important information: time scale, dosages, which organs were even examined?

l.248f: There are a couple of systems for MRI that are indeed coated or functionalized with PEG, but to say that MRI systems are generally coated with PEG is definitely not correct.

l.277f: A gold surface alone is not enough to make a particle system colloidally stable.

l.344f: Superparamagnetic materials don´t have a hysteresis and therefore ferromagnetic materials can be heated better than superparamagnets

Section 3.5.: No real discussion of the presented systems, e.g. in the MPI section it is not mentioned that MPI has to be combined with another imaging technique, e.g. CT, in order to locate the particles.

l.619ff: Despite the fact that volcanos and fires can cause serious risks to human health, what does this have to do with Nanoparticles?

l.643: As you already said at the beginning of your section, “dosis facit venenum”, i.e. you cannot state whether particles are toxic or not without considering dosage or concentration.

Section 4: Many aspects of SPION toxicity were not mentioned, e.g. CARPA reactions.

Author Response

We would like to thank the Reviewer for reviewing our manuscript and for pointing out to the issues  needing clarification or further explanation. 

We believe it will help us to improve our manuscript.

Round 2

Reviewer 1 Report

I can see improvement yet there are still multiple language mistakes and confusing statements, for example:

l. 37. The d unique? special? properties of SPIONs are their nanoscale sizes (usually in the range of 20 37 to 150 nm) and sensitivity to a magnetic field…. Instead of, for example: The small sizes () and sensitivity to a magnetic field make SPIONs unique… or so..

l. 89 Additional hematite (α-Fe2O3) nanoparticles … additionally, ??

l. 92. A transmission electron microscopy (TEM) measurements ..

l. 53: MRI contrast agent also needs a to be ..

l. 258: add a reference to a study on glioma using Fe3O4 labeled with for example: anti-IGFBP7 sdAbs..

l. 291: When a variable field B1 characterized by the frequency whose energy matches 291 that of separating the energy levels of spins is introduced.. confusing sentence..

l 259, 259.. “Registering the FID signal for protons in water molecules present in the tissue allows for creating its computer image; position information is obtained by using a spatial gradient of the magnetic field B0.”   Registering FID itself does not allow to obtain an MR image: applying variable magnetic field while exciting spins and acquiring MR signal allows 2D or 3D information on spin spatial distribution to be obtained and converted to an MR image using 2D FT…

l 302: “The MRI contrasts improve the resolution” – this is not true..

l. 312: first 7T whole body MRI system …  was recently ..

l 347: Finally, there is an increasing trend to use SPION properties in diagnostic and clinical techniques. The magnetic technique..   1) the properties are not used but SPIONs may be used; 2) what magnetic techniques the authors talk about here?

l. 443: All the efforts to introduce magnetic hyperthermia to clinical practice were completed in 2011; - this is not true, efforts were not completed. Researchers keep working in this area and, likely, will be working for long time…

l. 704: In contrast to MPI, CT (Computed Tomography) and MRI provide anatomical information… who/what provides this information?

Author Response

We would like to thank the Reviewer for all valuable comments.

Reviewer 3 Report

- Concerning the hematite particles with a Ms=3.98 emu/g (bzw, it is highly advised to use SI Units): This number is still significantly lower than Magnetite and maghemite formulations

- This whole part where the toxicity of QDs is discussed is absolutely not necessary, since they have nothing to do with SPIONs. There are some publications which discuss different surface effects, including charge, and their link to their toxicological Profile

- l.161ff: In this sentence you quote from Singh et al. that dextran coated SPIONs were cytotoxic at 50 µg/mL. But In the following you descirbe the results from Unterweger et al, which clearly show that dextran-coated SPION formulations clerly can be non toxic even at concentrations of 400 µg/mL. If you carefully read the section in Singh et al. Then you see that he describes the occurence of toxic effect, due to the brakedown of the dextran coating. As a result, the toxicity here is not arising due to the dextran shell, but due to the lack of stability of this shell. In Unterweger et al. it seems that the design the dextran coating is better since here no cytotoxic effects occur. In other words, not only the material but ist linkage to the particle surface Plays a critical role when it comes to stability and toxicity

Author Response

(The authors gave the same response as above.)
